# Adversarial Soft Advantage Fitting:
# Imitation Learning without Policy Optimization

**Paul Barde**[*†]
Québec AI institute (Mila)
McGill University
`bardepau@mila.quebec`

**Julien Roy**[*†]
Québec AI institute (Mila)
Polytechnique Montréal
`julien.roy@mila.quebec`

**Wonseok Jeon**[*]
Québec AI institute (Mila)
McGill University
`jeonwons@mila.quebec`

**Joelle Pineau**[‡]
Québec AI institute (Mila)
McGill University
Facebook AI Research

**Derek Nowrouzezahrai**
Québec AI institute (Mila)
McGill University

**Christopher Pal**[‡]
Québec AI institute (Mila)
Polytechnique Montréal
Element AI

## Abstract

Adversarial Imitation Learning alternates between learning a discriminator – which tells apart expert's demonstrations from generated ones – and a generator's policy to produce trajectories that can fool this discriminator. This alternated optimization is known to be delicate in practice since it compounds unstable adversarial training with brittle and sample-inefficient reinforcement learning. We propose to remove the burden of the policy optimization steps by leveraging a novel discriminator formulation. Specifically, our discriminator is explicitly conditioned on two policies: the one from the previous generator's iteration and a learnable policy. When optimized, this discriminator directly learns the optimal generator's policy. Consequently, our discriminator's update solves the generator's optimization problem for free: learning a policy that imitates the expert does not require an additional optimization loop. This formulation effectively cuts by half the implementation and computational burden of Adversarial Imitation Learning algorithms by removing the Reinforcement Learning phase altogether. We show on a variety of tasks that our simpler approach is competitive to prevalent Imitation Learning methods.

## 1   Introduction

Imitation Learning (IL) treats the task of learning a policy from a set of expert demonstrations. IL is effective on control problems that are challenging for traditional Reinforcement Learning (RL) methods, either due to reward function design challenges or the inherent difficulty of the task itself [1, 24].

Most IL work can be divided into two branches: Behavioral Cloning and Inverse Reinforcement Learning. Behavioral Cloning casts IL as a supervised learning objective and seeks to imitate the expert's actions using the provided demonstrations as a fixed dataset [19]. Thus, Behavioral Cloning usually requires a lot of expert data and results in agents that struggle to generalize. As an agent deviates from the demonstrated behaviors – straying outside the state distribution on which it was trained – the risks of making additional errors increase, a problem known as compounding error [24].

---

[*]Equal contribution.
[†]Work conducted while interning at Ubisoft Montreal's La Forge R&D laboratory.
[‡]Canada CIFAR AI Chair.

Inverse Reinforcement Learning aims to reduce compounding error by learning a reward function under which the expert policy is optimal [1]. Once learned, an agent can be trained (with any RL algorithm) to learn how to act at any given state of the environment. Early methods were prohibitively expensive on large environments because they required training the RL agent to convergence at each learning step of the reward function [30, 1]. Recent approaches instead apply an adversarial formulation (Adversarial Imitation Learning, AIL) in which a discriminator learns to distinguish between expert and agent behaviors to learn the reward optimized by the expert. AIL methods allow for the use of function approximators and can in practice be used with only a few policy improvement steps for each discriminator update [13, 6, 4].

While these advances have allowed Imitation Learning to tackle bigger and more complex environments [16, 3], they have also significantly complexified the implementation and learning dynamics of Imitation Learning algorithms. It is worth asking how much of this complexity is actually mandated. For example, in recent work, Reddy et al. [20] have shown that competitive performance can be obtained by hard-coding a very simple reward function to incentivize expert-like behaviors and manage to imitate it through off-policy direct RL. Reddy et al. [20] therefore remove the reward learning component of AIL and focus on the RL loop, yielding a regularized version of Behavioral Cloning. Motivated by these results, we also seek to simplify the AIL framework but following the opposite direction: keeping the reward learning module and removing the policy improvement loop.

We propose a simpler yet competitive AIL framework. Motivated by Finn et al. [4] who use the optimal discriminator form, we propose a structured discriminator that estimates the probability of demonstrated and generated behavior using a single parameterized maximum entropy policy. Discriminator learning and policy learning therefore occur simultaneously, rendering seamless generator updates: once the discriminator has been trained for a few epochs, we simply use its policy model to generate new rollouts. We call this approach Adversarial Soft Advantage Fitting (ASAF).

We make the following contributions:

- **Algorithmic**: we present a novel algorithm (ASAF) designed to imitate expert demonstrations without any Reinforcement Learning step.
- **Theoretical**: we show that our method retrieves the expert policy when trained to optimality.
- **Empirical**: we show that ASAF outperforms prevalent IL algorithms on a variety of discrete and continuous control tasks. We also show that, in practice, ASAF can be easily modified to account for different trajectory lengths (from full length to transition-wise).

## 2 Background

**Markov Decision Processes (MDPs)**  We use Hazan et al. [12]'s notation and consider the classic $T$-horizon $\gamma$-discounted MDP $\mathcal{M} = \langle \mathcal{S}, \mathcal{A}, \mathcal{P}, \mathcal{P}_0, \gamma, r, T \rangle$. For simplicity, we assume that $\mathcal{S}$ and $\mathcal{A}$ are finite. Successor states are given by the transition distribution $\mathcal{P}(s'|s,a) \in [0,1]$, and the initial state $s_0$ is drawn from $\mathcal{P}_0(s) \in [0,1]$. Transitions are rewarded with $r(s,a) \in \mathbb{R}$ with $r$ being bounded. The discount factor and the episode horizon are $\gamma \in [0,1]$ and $T \in \mathbb{N} \cup \{\infty\}$, where $T < \infty$ for $\gamma = 1$. Finally, we consider stationary stochastic policies $\pi \in \Pi : \mathcal{S} \times \mathcal{A} \to ]0,1[$ that produce trajectories $\tau = (s_0, a_0, s_1, a_1, ..., s_{T-1}, a_{T-1}, s_T)$ when executed on $\mathcal{M}$.

The probability of trajectory $\tau$ under policy $\pi$ is $P_\pi(\tau) \triangleq \mathcal{P}_0(s_0) \prod_{t=0}^{T-1} \pi(a_t|s_t) \mathcal{P}(s_{t+1}|s_t, a_t)$ and the corresponding marginals are defined as $d_{t,\pi}(s) \triangleq \sum_{\tau:s_t=s} P_\pi(\tau)$ and $d_{t,\pi}(s,a) \triangleq \sum_{\tau:s_t=s,a_t=a} P_\pi(\tau) = d_{t,\pi}(s)\pi(a|s)$, respectively. With these marginals, we define the normalized discounted state and state-action occupancy measures as $d_\pi(s) \triangleq \frac{1}{Z(\gamma,T)} \sum_{t=0}^{T-1} \gamma^t d_{t,\pi}(s)$ and $d_\pi(s,a) \triangleq \frac{1}{Z(\gamma,T)} \sum_{t=0}^{T-1} \gamma^t d_{t,\pi}(s,a) = d_\pi(s)\pi(a|s)$ where the partition function $Z(\gamma,T)$ is equal to $\sum_{t=0}^{T-1} \gamma^t$. Intuitively, the state (or state-action) occupancy measure can be interpreted as the discounted visitation distribution of the states (or state-action pairs) that the agent encounters when navigating with policy $\pi$. The expected sum of discounted rewards can be expressed in term of the occupancy measures as follows:

$$J_\pi[r(s,a)] \triangleq \mathbb{E}_{\tau \sim P_\pi} \left[ \sum_{t=0}^{T-1} \gamma^t r(s_t, a_t) \right] = Z(\gamma,T) \, \mathbb{E}_{(s,a) \sim d_\pi}[r(s,a)].$$

In the entropy-regularized Reinforcement Learning framework [11], the optimal policy maximizes its entropy at each visited state in addition to the standard RL objective:

$$\pi^* \triangleq \arg \max_{\pi} J_{\pi}[r(s,a) + \alpha \mathcal{H}(\pi(\cdot|s))], \quad \mathcal{H}(\pi(\cdot|s)) = \mathbb{E}_{a \sim \pi(\cdot|s)}[-\log(\pi(a|s))].$$

As shown in [29, 10] the corresponding optimal policy is

$$\pi^*_{\text{soft}}(a|s) = \exp\left(\alpha^{-1} A^*_{\text{soft}}(s,a)\right) \quad \text{with} \quad A^*_{\text{soft}}(s,a) \triangleq Q^*_{\text{soft}}(s,a) - V^*_{\text{soft}}(s), \tag{1}$$

$$V^*_{\text{soft}}(s) = \alpha \log \sum_{a \in \mathcal{A}} \exp\left(\alpha^{-1} Q^*_{\text{soft}}(s,a)\right), \; Q^*_{\text{soft}}(s,a) = r(s,a) + \gamma \mathbb{E}_{s' \sim \mathcal{P}(\cdot|s,a)}\left[V^*_{\text{soft}}(s')\right] \tag{2}$$

**Maximum Causal Entropy Inverse Reinforcement Learning**   In the problem of Inverse Reinforcement Learning (IRL), it is assumed that the MDP's reward function is unknown but that demonstrations from using expert's policy $\pi_E$ are provided. Maximum causal entropy IRL [30] proposes to fit a reward function $r$ from a set $\mathcal{R}$ of reward functions and retrieve the corresponding optimal policy by solving the optimization problem

$$\min_{r \in \mathcal{R}} \left(\max_{\pi} J_{\pi}[r(s,a) + \mathcal{H}(\pi(\cdot|s))]\right) - J_{\pi_E}[r(s,a)]. \tag{3}$$

In brief, the problem reduces to finding a reward function $r$ for which the expert policy is optimal. In order to do so, the optimization procedure searches high entropy policies that are optimal with respect to $r$ and minimizes the difference between their returns and the return of the expert policy, eventually reaching a policy $\pi$ that approaches $\pi_E$. Most of the proposed solutions [1, 29, 13] transpose IRL to the problem of distribution matching; Abbeel and Ng [1] and Ziebart et al. [30] used linear function approximation and proposed to match the feature expectation; Ho and Ermon [13] proposed to cast Eq. (3) with a convex reward function regularizer into the problem of minimizing the Jensen-Shannon divergence between the state-action occupancy measures:

$$\min_{\pi} D_{\text{JS}}(d_{\pi}, d_{\pi_E}) - J_{\pi}[\mathcal{H}(\pi(\cdot|s))] \tag{4}$$

**Connections between Generative Adversarial Networks (GANs) and IRL**   For the data distribution $p_E$ and the generator distribution $p_G$ defined on the domain $\mathcal{X}$, the GAN objective [8] is

$$\min_{p_G} \max_{D} L(D, p_G), \quad L(D, p_G) \triangleq \mathbb{E}_{x \sim p_E}[\log D(x)] + \mathbb{E}_{x \sim p_G}[\log(1 - D(x))]. \tag{5}$$

In Goodfellow et al. [8], the maximizer of the inner problem in Eq. (5) is shown to be

$$D^*_{p_G} \triangleq \arg \max_{D} L(D, p_G) = \frac{p_E}{p_E + p_G}, \tag{6}$$

and the optimizer for Eq. (5) is $\arg \min_{p_G} \max_D L(D, p_G) = \arg \min_{p_G} L(D^*_{p_G}, p_G) = p_E$. Later, Finn et al. [4] and Ho and Ermon [13] concurrently proposed connections between GANs and IRL. The Generative Adversarial Imitation Learning (GAIL) formulation in Ho and Ermon [13] is based on matching state-action occupancy measures, while Finn et al. [4] considered matching trajectory distributions. Our work is inspired by the discriminator proposed and used by Finn et al. [4],

$$D_{\theta}(\tau) \triangleq \frac{p_{\theta}(\tau)}{p_{\theta}(\tau) + q(\tau)}, \tag{7}$$

where $p_{\theta}(\tau) \propto \exp r_{\theta}(\tau)$ with reward approximator $r_{\theta}$ motivated by maximum causal entropy IRL. Note that Eq. (7) matches the form of the optimal discriminator in Eq. (6). Although Finn et al. [4] do not empirically support the effectiveness of their method, the Adversarial IRL approach of Fu et al. [6] (AIRL) successfully used a similar discriminator for state-action occupancy measure matching.

## 3   Imitation Learning without Policy Optimization

In this section, we derive Adversarial Soft Advantage Fitting (ASAF), our novel Adversarial Imitation Learning approach. Specifically, in Section 3.1, we present the theoretical foundations for ASAF to perform Imitation Learning on full-length trajectories. Intuitively, our method is based on the use

of such *structured discriminators* – that match the optimal discriminator form – to fit the trajectory distribution induced by the expert policy. This approach requires being able to evaluate and sample from the learned policy and allows us to learn that policy and train the discriminator simultaneously, thus drastically simplifying the training procedure. We present in Section 3.2 parametrization options that satisfy these requirements. Finally, in Section 3.3, we explain how to implement a practical algorithm that can be used for arbitrary trajectory-lengths, including the transition-wise case.

## 3.1 Adversarial Soft Advantage Fitting – Theoretical setting

Before introducing our method, we derive GAN training with a structured discriminator.

**GAN with structured discriminator**  Suppose that we have a generator distribution $p_G$ and some arbitrary distribution $\tilde{p}$ and that both can be evaluated efficiently, e.g., categorical distribution or probability density with normalizing flows [22]. We call a *structured discriminator* a function $D_{\tilde{p},p_G} : \mathcal{X} \to [0,1]$ of the form $D_{\tilde{p},p_G}(x) = \tilde{p}(x)/(\tilde{p}(x) + p_G(x))$ which matches the optimal discriminator form for Eq. (6). Considering our new GAN objective, we get:

$$\min_{p_G} \max_{\tilde{p}} L(\tilde{p}, p_G), \quad L(\tilde{p}, p_G) \triangleq \mathbb{E}_{x \sim p_E}[\log D_{\tilde{p},p_G}(x)] + \mathbb{E}_{x \sim p_G}[\log(1 - D_{\tilde{p},p_G}(x))]. \quad (8)$$

While the unstructured discriminator $D$ from Eq. (5) learns a mapping from $x$ to a Bernoulli distribution, we now learn a mapping from $x$ to an arbitrary distribution $\tilde{p}$ from which we can analytically compute $D_{\tilde{p},p_G}(x)$. One can therefore say that $D_{\tilde{p},p_G}$ is *parameterized* by $\tilde{p}$. For the optimization problem of Eq. (8), we have the following optima:

**Lemma 1.** *The optimal discriminator parameter for any generator $p_G$ in Eq. (8) is equal to the expert's distribution, $\tilde{p}^* \triangleq \arg\max_{\tilde{p}} L(\tilde{p}, p_G) = p_E$ , and the optimal discriminator parameter is also the optimal generator, i.e.,*

$$p_G^* \triangleq \arg\min_{p_G} \max_{\tilde{p}} L(\tilde{p}, p_G) = \arg\min_{p_G} L(p_E, p_G) = p_E = \tilde{p}^*.$$

*Proof.* See Appendix A.1

Intuitively, Lemma 1 shows that the optimal discriminator parameter is also the target data distribution of our optimization problem (i.e., the optimal generator). In other words, solving the inner optimization yields the solution of the outer optimization. In practice, we update $\tilde{p}$ to minimize the discriminator objective and use it directly as $p_G$ to sample new data.

**Matching trajectory distributions with structured discriminator**  Motivated by the GAN with structured discriminator, we consider the trajectory distribution matching problem in IL. Here, we optimise Eq. (8) with $x = \tau, \mathcal{X} = \mathcal{T}, p_E = P_{\pi_E}, p_G = P_{\pi_G}$, which yields the following objective:

$$\min_{\pi_G} \max_{\tilde{\pi}} L(\tilde{\pi}, \pi_G), \quad L(\tilde{\pi}, \pi_G) \triangleq \mathbb{E}_{\tau \sim P_{\pi_E}}[\log D_{\tilde{\pi},\pi_G}(\tau)] + \mathbb{E}_{\tau \sim P_{\pi_G}}[\log(1 - D_{\tilde{\pi},\pi_G}(\tau))], \quad (9)$$

with the structured discriminator:

$$D_{\tilde{\pi},\pi_G}(\tau) = \frac{P_{\tilde{\pi}}(\tau)}{P_{\tilde{\pi}}(\tau) + P_{\pi_G}(\tau)} = \frac{q_{\tilde{\pi}}(\tau)}{q_{\tilde{\pi}}(\tau) + q_{\pi_G}(\tau)}. \quad (10)$$

Here we used the fact that $P_\pi(\tau)$ decomposes into two distinct products: $q_\pi(\tau) \triangleq \prod_{t=0}^{T-1} \pi(a_t|s_t)$ which depends on the stationary policy $\pi$ and $\xi(\tau) \triangleq \mathcal{P}_0(s_0) \prod_{t=0}^{T-1} \mathcal{P}(s_{t+1}|s_t, a_t)$ which accounts for the environment dynamics. Crucially, $\xi(\tau)$ cancels out in the numerator and denominator leaving $\tilde{\pi}$ as the sole parameter of this structured discriminator. In this way, $D_{\tilde{\pi},\pi_G}(\tau)$ can evaluate the probability of a trajectory being generated by the expert policy simply by evaluating products of stationary policy distributions $\tilde{\pi}$ and $\pi_G$. With this form, we can get the following result:

**Theorem 1.** *The optimal discriminator parameter for any generator policy $\pi_G$ in Eq. (9) $\tilde{\pi}^* \triangleq \arg\max_{\tilde{\pi}} L(\tilde{\pi}, \pi_G)$ is such that $q_{\tilde{\pi}^*} = q_{\pi_E}$, and using generator policy $\tilde{\pi}^*$ minimizes $L(\tilde{\pi}^*, \pi_G)$, i.e.,*

$$\tilde{\pi}^* \in \arg\min_{\pi_G} \max_{\tilde{\pi}} L(\tilde{\pi}, \pi_G) = \arg\min_{\pi_G} L(\tilde{\pi}^*, \pi_G).$$

*Proof.* See Appendix A.2

Theorem 1's benefits are similar to the ones from Lemma 1: we can use a discriminator of the form of Eq. (10) to fit to the expert demonstrations a policy $\tilde{\pi}^*$ that simultaneously yields the optimal generator's policy and produces the same trajectory distribution as the expert policy.

## 3.2   A Specific Policy Class

The derivations of Section 3.1 rely on the use of a learnable policy that can both be evaluated and sampled from in order to fit the expert policy. A number of parameterization options that satisfy these conditions are available.

First of all, we observe that since $\pi_E$ is independent of $r$ and $\pi$, we can add the entropy of the expert policy $\mathcal{H}(\pi_E(\cdot|s))$ to the MaxEnt IRL objective of Eq. (3) without modifying the solution to the optimization problem:

$$\min_{r \in \mathcal{R}} \left( \max_{\pi \in \Pi} J_{\pi}[r(s,a) + \mathcal{H}(\pi(\cdot|s))] \right) - J_{\pi_E}[r(s,a) + \mathcal{H}(\pi_E(\cdot|s))] \tag{11}$$

The max over policies implies that when optimising $r$, $\pi$ has already been made optimal with respect to the causal entropy augmented reward function $r'(s,a|\pi) = r(s,a) + \mathcal{H}(\pi(\cdot|s))$ and therefore it must be of the form presented in Eq. (1). Moreover, since $\pi$ is optimal w.r.t. $r'$ the difference in performance $J_{\pi}[r'(s,a|\pi)] - J_{\pi_E}[r'(s,a|\pi_E)]$ is always non-negative and its minimum of 0 is only reached when $\pi_E$ is also optimal w.r.t. $r'$, in which case $\pi_E$ must also be of the form of Eq. (1).

With discrete action spaces we propose to parameterize the MaxEnt policy defined in Eq. (1) with the following categorical distribution $\tilde{\pi}(a|s) = \exp\left(Q_\theta(s,a) - \log\sum_{a'}\exp Q_\theta(s,a')\right)$, where $Q_\theta$ is a model parameterized by $\theta$ that approximates $\frac{1}{\alpha}Q^*_{\text{soft}}$.

With continuous action spaces, the soft value function involves an intractable integral over the action domain. Therefore, we approximate the MaxEnt distribution with a Normal distribution with diagonal covariance matrix like it is commonly done in the literature [11, 17]. By parameterizing the mean and variance we get a learnable density function that can be easily evaluated and sampled from.

## 3.3   Adversarial Soft Advantage Fitting (ASAF) – practical algorithm

Section 3.1 shows that assuming $\tilde{\pi}$ can be evaluated and sampled from, we can use the structured discriminator of Eq. (10) to learn a policy $\tilde{\pi}$ that matches the expert's trajectory distribution. Section 3.2 proposes parameterizations for discrete and continuous action spaces that satisfy those assumptions.

In practice, as with GANs [8], we do not train the discriminator to convergence as gradient-based optimisation cannot be expected to find the global optimum of non-convex problems. Instead, Adversarial Soft Advantage Fitting (ASAF) alternates between two simple steps: (1) training $D_{\tilde{\pi},\pi_G}$ by minimizing the binary cross-entropy loss,

$$\mathcal{L}_{BCE}(\mathcal{D}_E, \mathcal{D}_G, \tilde{\pi}) \approx -\frac{1}{n_E}\sum_{i=1}^{n_E} \log D_{\tilde{\pi},\pi_G}(\tau_i^{(E)}) - \frac{1}{n_G}\sum_{i=1}^{n_G} \log\left(1 - D_{\tilde{\pi},\pi_G}(\tau_i^{(G)})\right)$$

$$\text{where} \quad \tau_i^{(E)} \sim \mathcal{D}_E \,, \tau_i^{(G)} \sim \mathcal{D}_G \text{ and } D_{\tilde{\pi},\pi_G}(\tau) = \frac{\prod_{t=0}^{T-1}\tilde{\pi}(a_t|s_t)}{\prod_{t=0}^{T-1}\tilde{\pi}(a_t|s_t) + \prod_{t=0}^{T-1}\pi_G(a_t|s_t)} \tag{12}$$

with minibatch sizes $n_E = n_G$, and (2) updating the generator's policy as $\pi_G \leftarrow \tilde{\pi}$ to minimize Eq. (9) (see Algorithm 1).

We derived ASAF considering full trajectories, yet it might be preferable in practice to split full trajectories into smaller chunks. This is particularly true in environments where trajectory length varies a lot or tends to infinity.

To investigate whether the practical benefits of using partial trajectories hurt ASAF's performance, we also consider a variation, ASAF-$w$, where we treat trajectory-windows of size $w$ as if they were full trajectories. Note that considering windows as full trajectories results in approximating that the initial state of these sub-trajectories have equal probability under the expert's and the generator's policy (this is easily seen when deriving Eq. (10)).

In the limit, ASAF-1 (window-size of 1) becomes a transition-wise algorithm which can be desirable if one wants to collect rollouts asynchronously or has only access to unsequential expert data. While ASAF-1 may work well in practice it essentially assumes that the expert's and the generator's policies have the same state occupancy measure, which is incorrect until actually recovering the true expert policy.

---

**Algorithm 1:** ASAF

---

**Require:** expert trajectories $\mathcal{D}_E = \{\tau_i\}_{i=1}^{N_E}$

Randomly initialize $\tilde{\pi}$ and set $\pi_G \leftarrow \tilde{\pi}$

**for** steps $m = 0$ to $M$ **do**

    Collect trajectories $\mathcal{D}_G = \{\tau_i\}_{i=1}^{N_G}$ using $\pi_G$

    Update $\tilde{\pi}$ by minimizing Eq. (12)

    Set $\pi_G \leftarrow \tilde{\pi}$

**end for**

---

Finally, to offer a complete family of algorithms based on the structured discriminator approach, we show in Appendix B that this assumption is not mandatory and derive a transition-wise algorithm based on Soft Q-function Fitting (rather than soft advantages) that also gets rid of the RL loop. We call this algorithm ASQF. While theoretically sound, we found that in practice, ASQF is outperformed by ASAF-1 in more complex environments (see Section 5.1).

## 4   Related works

Ziebart et al. [30] first proposed MaxEnt IRL, the foundation of modern IL. Ziebart [29] further elaborated MaxEnt IRL as well as deriving the optimal form of the MaxEnt policy at the core of our methods. Finn et al. [4] proposed a GAN formulation to IRL that leveraged the energy based models of Ziebart [29]. Finn et al. [5]'s implementation of this method, however, relied on processing full trajectories with Linear Quadratic Regulator and on optimizing with guided policy search, to manage the high variance of trajectory costs. To retrieve robust rewards, Fu et al. [6] proposed a straightforward transposition of [4] to state-action transitions. In doing so, they had to however do away with a GAN objective during policy optimization, consequently minimizing the Kullback–Leibler divergence from the expert occupancy measure to the policy occupancy measure (instead of the Jensen-Shannon divergence) [7].

Later works [25, 15] move away from the Generative Adversarial formulation. To do so, Sasaki et al. [25] directly express the expectation of the Jensen-Shannon divergence between the occupancy measures in term of the agent's Q-function, which can then be used to optimize the agent's policy with off-policy Actor-Critic [2]. Similarly, Kostrikov et al. [15] use Dual Stationary Distribution Correction Estimation [18] to approximate the Q-function on the expert's demonstrations before optimizing the agent's policy under the initial state distribution using the reparametrization trick [11]. While [25, 15] are related to our methods in their interests in learning directly the value function, they differ in their goal and thus in the resulting algorithmic complexity. Indeed, they aim at improving the sample efficiency in terms of environment interaction and therefore move away from the algorithmically simple Generative Adversarial formulation towards more complicated divergence minimization methods. In doing so, they further complicate the Imitation Learning methods while still requiring to explicitly learn a policy. Yet, simply using the Generative Adversarial formulation with an Experience Replay Buffer can significantly improve the sample efficiency [14]. For these reasons, and since our aim is to propose efficient yet simple methods, we focus on the Generative Adversarial formulation.

While Reddy et al. [20] share our interest for simpler IL methods, they pursue an opposite approach to ours. They propose to eliminate the reward learning steps of IRL by simply hard-coding a reward of 1 for expert's transitions and of 0 for agent's transitions. They then use Soft Q-learning [10] to learn a value function by sampling transitions in equal proportion from the expert's and agent's buffers. Unfortunately, once the learner accurately mimics the expert, it collects expert-like transitions that are labeled with a reward of 0 since they are generated and not coming from the demonstrations. This effectively causes the reward of expert-like behavior to decay as the agent improves and can severely destabilize learning to a point where early-stopping becomes required [20].

Our work builds on [4], yet its novelty is to explicitly express the probability of a trajectory in terms of the policy in order to directly learn this latter when training the discriminator. In contrast, [6] considers a transition-wise discriminator with un-normalized probabilities which makes it closer to ASQF (Appendix B) than to ASAF-1. Additionally, AIRL [6] minimizes the Kullback-Leiber Divergence [7] between occupancy measures whereas ASAF minimizes the Jensen-Shanon Divergence between trajectory distributions.

Finally, Behavioral Cloning uses the loss function from supervised learning (classification or regression) to match expert's actions given expert's states and suffers from compounding error due to co-variate shift [23] since its data is limited to the demonstrated state-action pairs without environment interaction. Contrarily, ASAF-1 uses the binary cross entropy loss in Eq. (12) and does not suffer from compounding error as it learns on both generated and expert's trajectories.

## 5 Results and discussion

We evaluate our methods on a variety of discrete and continuous control tasks. Our results show that, in addition to drastically simplifying the adversarial IRL framework, our methods perform on par or better than previous approaches on all but one environment. When trajectory length is really long or drastically varies across episodes (see MuJoCo experiments Section 5.3), we find that using sub-trajectories with fixed window-size (ASAF-$w$ or ASAF-1) significantly outperforms its full trajectory counterpart ASAF.

### 5.1 Experimental setup

We compare our algorithms ASAF, ASAF-$w$ and ASAF-1 against GAIL [13], the predominant Adversarial Imitation Learning algorithm in the litterature, and AIRL [6], one of its variations that also leverages the access to the generator's policy distribution. Additionally, we compare against SQIL [20], a recent Reinforcement Learning-only approach to Imitation Learning that proved successful on high-dimensional tasks. Our implementations of GAIL and AIRL use PPO [27] instead of TRPO [26] as it has been shown to improve performance [14]. Finally, to be consistent with [13], we do not use causal entropy regularization.

For all tasks except MuJoCo, we selected the best performing hyperparameters through a random search of equal budget for each algorithm-environment pair (see Appendix D) and the best configuration is retrained on ten random seeds. For the MuJoCo experiments, GAIL required extensive tuning (through random searches) of both its RL and IRL components to achieve satisfactory performances. Our methods, ASAF-$w$ and ASAF-1, on the other hand showed much more stable and robust to hyperparameterization, which is likely due to their simplicity. SQIL used the same SAC[11] implementation and hyperparameters that were used to generate the expert demonstrations.

Finally for each task, all algorithms use the same neural network architectures for their policy and/or discriminator (see full description in Appendix D). Expert demonstrations are either generated by hand (mountaincar), using open-source bots (Pommerman) or from our implementations of SAC and PPO (all remaining). More details are given in Appendix E.

### 5.2 Experiments on classic control and Box2D tasks (discrete and continuous)

Figure 1 shows that ASAF and its approximate variations ASAF-1 and ASAF-$w$ quickly converge to expert's performance (here $w$ was tuned to values between 32 to 200, see Appendix D for selected window-sizes). This indicates that the practical benefits of using shorter trajectories or even just transitions does not hinder performance on these simple tasks. Note that for Box2D and classic control environments, we retrain the best configuration of each algorithm for twice as long than was done in the hyperparameter search, which allows to uncover unstable learning behaviors. Figure 1 shows that our methods display much more stable learning: their performance rises until they match the expert's and does not decrease once it is reached. This is a highly desirable property for an Imitation Learning algorithm since in practice one does not have access to a reward function and thus cannot monitor the performance of the learning algorithm to trigger early-stopping. The baselines on the other hand experience occasional performance drops. For GAIL and AIRL, this is likely due to the concurrent RL and IRL loops, whereas for SQIL, it has been noted that an effective reward decay can occur when accurately mimicking the expert [20]. This instability is particularly severe in the continuous control case. In practice, all three baselines use early stopping to avoid performance decay [20].

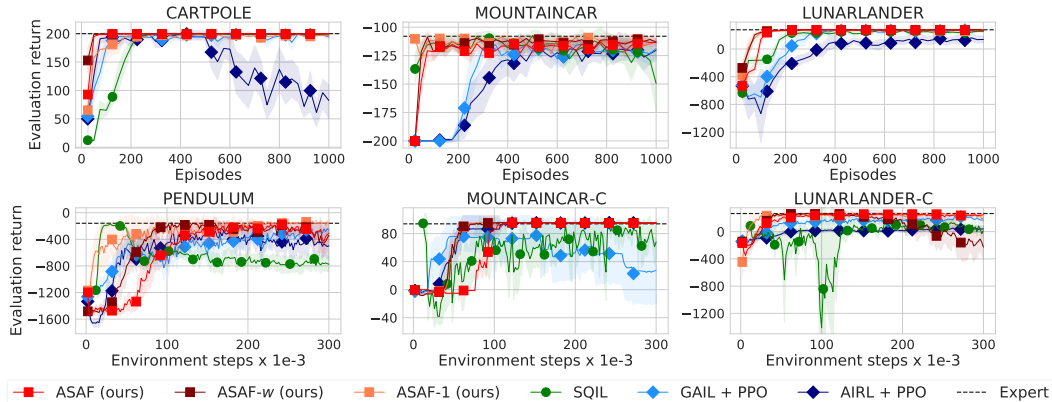

Figure 1: Results on classic control and Box2D tasks for 10 expert demonstrations. First row contains discrete actions environments, second row corresponds to continuous control.

## 5.3 Experiments on MuJoCo (continuous control)

To scale up our evaluations in continuous control we use the popular MuJoCo benchmarks. In this domain, the trajectory length is either fixed at a large value (1000 steps on HalfCheetah) or varies a lot across episodes due to termination when the character falls down (Hopper, Walker2d and Ant). Figure 2 shows that these trajectory characteristics hinder ASAF's learning as ASAF requires collecting multiple episodes for every update, while ASAF-1 and ASAF-*w* perform well and are more sample-efficient than ASAF in these scenarios. We focus on GAIL since [6] claim that AIRL performs on par with it on MuJoCo environments. In Figure 5 in Appendix C we evaluate GAIL both with and without gradient penalty (GP) on discriminator updates [9, 14] and while GAIL was originally proposed without GP [13], we empirically found that GP prevents the discriminator to overfit and enables RL to exploit dense rewards, which highly improves its sample efficiency. Despite these ameliorations, GAIL proved to be quite inconsistent across environments despite substantial efforts on hyperparameter tuning. On the other hand, ASAF-1 performs well across all environments. Finally, we see that SQIL's instability is exacerbated on MuJoCo.

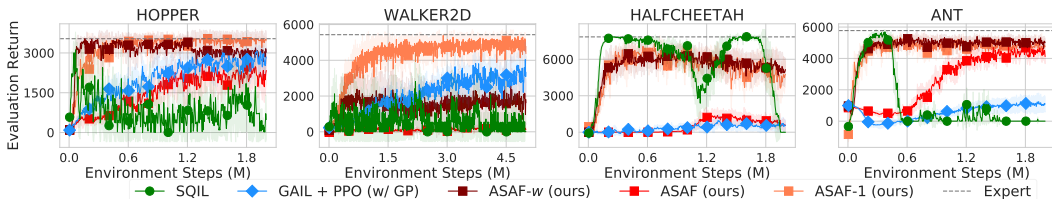

Figure 2: Results on MuJoCo tasks for 25 expert demonstrations.

## 5.4 Experiments on Pommerman (discrete control)

Finally, to scale up our evaluations in discrete control environments, we consider the domain of Pommerman [21], a challenging and very dynamic discrete control environment that uses rich and high-dimensional observation spaces (see Appendix E). We perform evaluations of all of our methods and baselines on a 1 vs 1 task where a learning agent plays against a random agent, the opponent. The goal for the learning agent is to navigate to the opponent and eliminate it using expert demonstrations provided by the champion algorithm of the FFA 2018 competition [28]. We removed the ability of the opponent to lay bombs so that it doesn't accidentally eliminate itself. Since it can still move around, it is however surprisingly tricky to eliminate: the expert has to navigate across the whole map, lay a bomb next to the opponent and retreat to avoid eliminating itself. This entire routine has then to be repeated several times until finally succeeding since the opponent will often avoid the hit by chance. We refer to this task as *Pommerman Random-Tag*. Note that since we measure success of the imitation task with the win-tie-lose outcome (sparse performance metric), a learning agent has to truly reproduce the expert behavior until the very end of trajectories to achieve higher scores.

Figure 3 shows that all three variations of ASAF as well as Behavioral Cloning (BC) outperform the baselines.

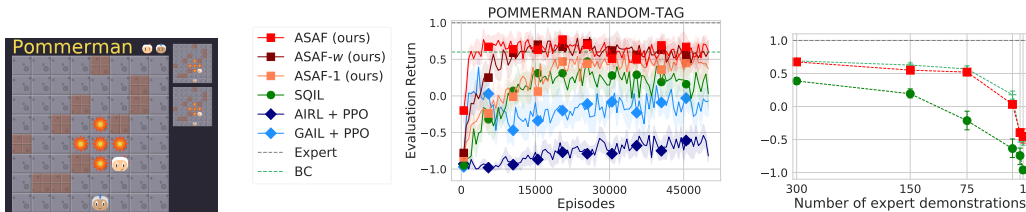

Figure 3: Results on Pommerman Random-Tag: (Left) Snapshot of the environment. (Center) Learning measured as evaluation return over episodes for 150 expert trajectories (Right) Average return on last 20% of training for decreasing number of expert trajectories [300, 150, 75, 15, 5, 1].

## 6   Conclusion

We propose an important simplification to the Adversarial Imitation Learning framework by removing the Reinforcement Learning optimisation loop altogether. We show that, by using a particular form for the discriminator, our method recovers a policy that matches the expert's trajectory distribution. We evaluate our approach against prior works on many different benchmarking tasks and show that our method (ASAF) compares favorably to the predominant Imitation Learning algorithms. The approximate versions, ASAF-*w* and ASAF-1, that use sub-trajectories yield a flexible algorithms that work well both on short and long time horizons. Finally, our approach still involves a reward learning module through its discriminator, and it would be interesting in future work to explore how ASAF can be used to learn robust rewards, along the lines of Fu et al. [6].

## Broader Impact

Our contributions are mainly theoretical and aim at simplifying current Imitation Learning methods. We do not propose new applications nor use sensitive data or simulator. Yet our method can ease and promote the use, design and development of Imitation Learning algorithms and may eventually lead to applications outside of simple and controlled simulators. We do not pretend to discuss the ethical implications of the general use of autonomous agents but we rather try to investigate what are some of the differences in using Imitation Learning rather than reward oriented methods in the design of such agents.

Using only a scalar reward function to specify the desired behavior of an autonomous agent is a challenging task as one must weight different desiderata and account for unsuspected behaviors and situations. Indeed, it is well known in practice that Reinforcement Learning agents tend to find bizarre ways of exploiting the reward signal without solving the desired task. The fact that it is difficult to specify and control the behavior or an RL agents is a major flaw that prevent current methods to be applied to risk sensitive situations. On the other hand, Imitation Learning proposes a more natural way of specifying nuanced preferences by demonstrating desirable ways of solving a task. Yet, IL also has its drawbacks. First of all one needs to be able to demonstrate the desired behavior and current methods tend to be only as good as the demonstrator. Second, it is a challenging problem to ensure that the agent will be able to adapt to new situations that do not resemble the demonstrations. For these reasons, it is clear for us that additional safeguards are required in order to apply Imitation Learning (and Reinforcement Learning) methods to any application that could effectively have a real world impact.

## Acknowledgments and Disclosure of Funding

We thank Eloi Alonso, Olivier Delalleau, Félix G. Harvey, Maxim Peter and the entire research team at Ubisoft Montreal's La Forge R&D laboratory. Their feedback and comments contributed significantly to this work. Christopher Pal and Derek Nowrouzezahrai acknowledge funding from the Fonds de Recherche Nature et Technologies (FRQNT), Ubisoft Montreal and Mitacs' Accelerate

Program in support of our work, as well as Compute Canada for providing computing resources. Derek and Paul also acknowledge support from the NSERC Industrial Research Chair program.

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
