[Supplementary Material]

# Appendix

## A Proofs

### A.1 Proof of Lemma 1

*Proof.* Lemma 1 states that given $L(\tilde{p}, p_G)$ defined in Eq. (8):

(a) $\tilde{p}^* \triangleq \arg\max_{\tilde{p}} L(\tilde{p}, p_G) = p_E$

(b) $\arg\min_{p_G} L(p_E, p_G) = p_E$

Starting with (a), we have:

$$\arg\max_{\tilde{p}} L(\tilde{p}, p_G) = \arg\max_{\tilde{p}} \sum_{x_i} p_E(x_i) \log D_{\tilde{p}, p_G}(x_i) + p_G(x_i) \log(1 - D_{\tilde{p}, p_G}(x_i))$$

$$\triangleq \arg\max_{\tilde{p}} \sum_{x_i} L_i$$

Assuming infinite discriminator's capacity, $L_i$ can be made independent for all $x_i \in \mathcal{X}$ and we can construct our optimal discriminator $D_{\tilde{p}, p_G}^*$ as a look-up table $D_{\tilde{p}, p_G}^* : \mathcal{X} \to ]0, 1[ \; ; \; x_i \mapsto D_i^*$ with $D_i^*$ the optimal discriminator for each $x_i$ defined as:

$$D_i^* = \arg\max_{D_i} L_i = \arg\max_{D_i} p_{E,i} \log D_i + p_{G,i} \log(1 - D_i), \tag{13}$$

with $p_{G,i} \triangleq p_G(x_i)$, $p_{E,i} \triangleq p_E(x_i)$ and $D_i \triangleq D(x_i)$.

Recall that $D_i \in ]0, 1[$ and that $p_{G,i} \in ]0, 1[$. Therefore the function $\tilde{p}_i \mapsto D_i = \dfrac{\tilde{p}_i}{\tilde{p}_i + p_{G,i}}$ is defined for $\tilde{p}_i \in ]0, +\infty[$. Since it is strictly monotonic over that domain we have that:

$$D_i^* = \arg\max_{D_i} L_i \; \Leftrightarrow \; \tilde{p}_i^* = \arg\max_{\tilde{p}_i} L_i \tag{14}$$

Taking the derivative and setting to zero, we get:

$$\left. \frac{dL_i}{d\tilde{p}_i} \right|_{\tilde{p}_i} = 0 \; \Leftrightarrow \; \tilde{p}_i = p_{E,i} \tag{15}$$

The second derivative test confirms that we have a maximum, i.e. $\left. \dfrac{d^2 L_i}{d\tilde{p}_i^2} \right|_{\tilde{p}_i^*} < 0$. The values of $L_i$ at the boundaries of the domain of definition of $\tilde{p}_i$ tend to $-\infty$, therefore $L_i(\tilde{p}_i^* = p_{E,i})$ is the global maximum of $L_i$ w.r.t. $\tilde{p}_i$. Finally, the optimal global discriminator is given by:

$$D_{\tilde{p}, p_G}^*(x) = \frac{p_E(x)}{p_E(x) + p_G(x)} \quad \forall x \in \mathcal{X} \tag{16}$$

This concludes the proof for (a).

The proof for (b) can be found in the work of Goodfellow et al. [8]. We reproduce it here for completion. Since from (a) we know that $\tilde{p}^*(x) = p_E(x) \, \forall x \in \mathcal{X}$, we can write the GAN objective for the optimal discriminator as:

$$\arg\min_{p_G} L(\tilde{p}^*, p_G) = \arg\min_{p_G} L(p_E, p_G) \tag{17}$$

$$= \arg\min_{p_G} \mathbb{E}_{x \sim p_E} \left[ \log \frac{p_E(x)}{p_E(x) + p_G(x)} \right] + \mathbb{E}_{x \sim p_G} \left[ \log \frac{p_G(x)}{p_E(x) + p_G(x)} \right] \tag{18}$$

Note that:

$$\log 4 = \mathbb{E}_{x \sim p_E} \left[ \log 2 \right] + \mathbb{E}_{x \sim p_G} \left[ \log 2 \right] \tag{19}$$

Adding Eq. (19) to Eq. (18) and subtracting $\log 4$ on both sides:

$$\arg\min_{p_G} L(p_E, p_G) = -\log 4 + \mathbb{E}_{x \sim p_E}\left[\log \frac{2p_E(x)}{p_E(x) + p_G(x)}\right] + \mathbb{E}_{x \sim p_G}\left[\log \frac{2p_G(x)}{p_E(x) + p_G(x)}\right] \tag{20}$$

$$= -\log 4 + D_{\mathrm{KL}}\left(p_E \left\|\frac{p_E + p_G}{2}\right.\right) + D_{\mathrm{KL}}\left(p_E \left\|\frac{p_E + p_G}{2}\right.\right) \tag{21}$$

$$= -\log 4 + 2D_{\mathrm{JS}}\left(p_E \| p_G\right) \tag{22}$$

Where $D_{\mathrm{KL}}$ and $D_{\mathrm{JS}}$ are respectively the Kullback-Leibler and the Jensen-Shannon divergences. Since the Jensen-Shannon divergence between two distributions is always non-negative and zero if and only if the two distributions are equal, we have that $\arg\min_{p_G} L(p_E, p_G) = p_E$.

This concludes the proof for (b). □

## A.2 Proof of Theorem 1

*Proof.* Theorem 1 states that given $L(\tilde{\pi}, \pi_G)$ defined in Eq. (9):

(a) $\tilde{\pi}^* \triangleq \arg\max_{\tilde{\pi}} L(\tilde{\pi}, \pi_G)$ satisfies $q_{\tilde{\pi}^*} = q_{\pi_E}$

(b) $\pi_G^* = \tilde{\pi}^* \in \arg\min_{\pi_G} L(\tilde{\pi}^*, \pi_G)$

The proof of (a) is very similar to the one from Lemma 1. Starting from Eq. (9) we have:

$$\arg\max_{\tilde{\pi}} L(\tilde{\pi}, \pi_G) = \arg\max_{\tilde{\pi}} \sum_{\tau_i} P_{\pi_E}(\tau_i) \log D_{\tilde{\pi}, \pi_G}(\tau_i) + P_{\pi_G}(\tau_i) \log(1 - D_{\tilde{\pi}, \pi_G}(\tau_i)) \tag{23}$$

$$= \arg\max_{\tilde{\pi}} \sum_{\tau_i} \xi(\tau_i)\left(q_{\pi_E}(\tau_i) \log D_{\tilde{\pi}, \pi_G}(\tau_i) + q_{\pi_G}(\tau_i) \log(1 - D_{\tilde{\pi}, \pi_G}(\tau_i))\right) \tag{24}$$

$$= \arg\max_{\tilde{\pi}} \sum_{\tau_i} L_i \tag{25}$$

Like for Lemma 1, we can optimise for each $L_i$ individually. When doing so, $\xi(\tau_i)$ can be omitted as it is constant w.r.t $\tilde{\pi}$. The rest of the proof is identical to the one of but Lemma 1 with $p_E = q_{\pi_E}$ and $p_G = q_{\pi_G}$. It follows that the max of $L(\tilde{\pi}, \pi_G)$ is reached for $q_{\tilde{\pi}}^* = q_{\pi_E}$. From that we obtain that the policy $\tilde{\pi}^*$ that makes the discriminator $D_{\tilde{\pi}^*, \pi_G}$ optimal w.r.t $L(\tilde{\pi}, \pi_G)$ is such that $q_{\tilde{\pi}^*} = q_{\tilde{\pi}}^* = q_{\pi_E}$ i.e. $\prod_{t=0}^{T-1} \tilde{\pi}^*(a_t|s_t) = \prod_{t=0}^{T-1} \pi_E(a_t|s_t) \forall \tau$.

The proof for (b) stems from the observation that choosing $\pi_G = \tilde{\pi}^*$ (the policy recovered by the optimal discriminator $D_{\tilde{\pi}^*, \pi_G}$) minimizes $L(\tilde{\pi}^*, \pi_G)$:

$$\pi_G(a|s) = \tilde{\pi}^*(a|s) \ \forall (s,a) \in \mathcal{S} \times \mathcal{A} \ \Rightarrow \ \prod_{t=0}^{T-1} \pi_G(a_t|s_t) = \prod_{t=0}^{T-1} \tilde{\pi}^*(a_t|s_t) \ \forall \tau \in \mathcal{T} \tag{26}$$

$$\Rightarrow \ q_{\pi_G}(\tau) = q_{\pi_E}(\tau) \ \forall \tau \in \mathcal{T} \tag{27}$$

$$\Rightarrow \ D_{\tilde{\pi}^*, \tilde{\pi}^*} = \frac{1}{2} \ \forall \tau \in \mathcal{T} \tag{28}$$

$$\Rightarrow \ L(\tilde{\pi}^*, \tilde{\pi}^*) = -\log 4 \tag{29}$$

By multiplying the numerator and denominator of $D_{\tilde{\pi}^*, \tilde{\pi}^*}$ by $\xi(\tau)$ it can be shown in exactly the same way as in Appendix A.1 that $-\log 4$ is the global minimum of $L(\tilde{\pi}^*, \pi_G)$. □

# B Adversarial Soft Q-Fitting: transition-wise Imitation Learning without Policy Optimization

In this section we present Adversarial Soft Q-Fitting (ASQF), a principled approach to Imitation Learning without Reinforcement Learning that relies exclusively on transitions. Using transitions rather than trajectories presents several practical benefits such as the possibility to deal with asynchronously collected data or non-sequential experts demonstrations. We first present the theoretical setting for ASQF and then test it on a variety of discrete control tasks. We show that while it is theoretically sound, ASQF is often outperformed by ASAF-1, an approximation to ASAF that also allows to rely on transitions instead of trajectories.

**Theoretical Setting** We consider the GAN objective of Eq. (5) with $x = (s, a)$, $\mathcal{X} = \mathcal{S} \times \mathcal{A}$, $p_E = d_{\pi_E}$, $p_G = d_{\pi_G}$ and a discriminator $D_{\tilde{f}, \pi_G}$ of the form of Fu et al. [6]:

$$\min_{\pi_G} \max_{\tilde{f}} L(\tilde{f}, \pi_G), \quad L(\tilde{f}, \pi_G) \triangleq \mathbb{E}_{d_{\pi_E}}[\log D_{\tilde{f}, \pi_G}(s, a)] + \mathbb{E}_{d_{\pi_G}}[\log(1 - D_{\tilde{f}, \pi_G}(s, a))],$$

$$\text{with} \quad D_{\tilde{f}, \pi_G} = \frac{\exp \tilde{f}(s, a)}{\exp \tilde{f}(s, a) + \pi_G(a|s)}, \tag{30}$$

for which we present the following theorem.

**Theorem 2.** *For any generator policy $\pi_G$, the optimal discriminator parameter for Eq. (30) is*

$$\tilde{f}^* \triangleq \arg\max_{\tilde{f}} L(\tilde{f}, \pi_G) = \log\left(\pi_E(a|s) \frac{d_{\pi_E}(s)}{d_{\pi_G}(s)}\right) \quad \forall (s, a) \in \mathcal{S} \times \mathcal{A}$$

*Using $\tilde{f}^*$, the optimal generator policy $\pi_G^*$ is*

$$\arg\min_{\pi_G} \max_{\tilde{f}} L(\tilde{f}, \pi_G) = \arg\min_{\pi_G} L(\tilde{f}^*, \pi_G) = \pi_E(a|s) = \frac{\exp \tilde{f}^*(s, a)}{\sum_{a'} \exp \tilde{f}^*(s, a')} \quad \forall (s, a) \in \mathcal{S} \times \mathcal{A}.$$

*Proof.* The beginning of the proof closely follows the proof of Appendix A.1.

$$\arg\max_{\tilde{f}} L(\tilde{f}, \pi_G) =$$

$$\arg\max_{\tilde{f}} \sum_{s_i, a_i} d_{\pi_E}(s_i, a_i) \log D_{\tilde{f}, \pi_G}(s_i, a_i) + d_{\pi_G}(s_i, a_i) \log(1 - D_{\tilde{f}, \pi_G}(s_i, a_i)) \tag{31}$$

We solve for each individual $(s_i, a_i)$ pair and note that $\tilde{f}_i \mapsto D_i = \dfrac{\exp \tilde{f}_i}{\exp \tilde{f}_i + \pi_{G,i}}$ is strictly monotonic on $\tilde{f}_i \in \mathbb{R} \, \forall \, \pi_{G,i} \in ]0, 1[$ so,

$$D_i^* = \arg\max_{D_i} L_i \Leftrightarrow \tilde{f}_i^* = \arg\max_{\tilde{f}} L_i \tag{32}$$

Taking the derivative and setting it to 0, we find that

$$\left. \frac{dL_i}{d\tilde{f}_i} \right|_{\tilde{f}_i} = 0 \quad \Leftrightarrow \quad \tilde{f}_i = \log\left(\pi_{G,i} \frac{d_{\pi_E, i}}{d_{\pi_G, i}}\right) \tag{33}$$

We confirm that we have a global maximum with the second derivative test and the values at the border of the domain i.e. $\left. \dfrac{d^2 L_i}{d\tilde{f}_i^2} \right|_{\tilde{f}_i^*} < 0$ and $L_i$ goes to $-\infty$ for $\tilde{f}_i \to +\infty$ and for $\tilde{f}_i \to -\infty$.

It follows that

$$\tilde{f}^*(s,a) = \log\left(\pi_G(a|s)\frac{d_{\pi_E}(s,a)}{d_{\pi_G}(s,a)}\right) \quad \forall (s,a) \in \mathcal{S} \times \mathcal{A} \tag{34}$$

$$\implies \tilde{f}^*(s,a) = \log\left(\pi_G(a|s)\frac{d_{\pi_E}(s)\pi_E(a|s)}{d_{\pi_G}(s)\pi_G(a|s)}\right) \quad \forall (s,a) \in \mathcal{S} \times \mathcal{A} \tag{35}$$

$$\implies \tilde{f}^*(s,a) = \log\left(\pi_E(a|s)\frac{d_{\pi_E}(s)}{d_{\pi_G}(s)}\right) \quad \forall (s,a) \in \mathcal{S} \times \mathcal{A} \tag{36}$$

This proves the first part of Theorem 2.

To prove the second part notice that

$$
\begin{aligned}
D_{\tilde{f}^*,\pi_G}(s,a) &= \frac{\pi_E(a|s)\dfrac{d_{\pi_E}(s)}{d_{\pi_G}(s)}}{\pi_E(a|s)\dfrac{d_{\pi_E}(s)}{d_{\pi_G}(s)} + \pi_G(a|s)} \\
&= \frac{\pi_E(a|s)d_{\pi_E}(s)}{\pi_E(a|s)d_{\pi_E}(s) + \pi_G(a|s)d_{\pi_G}(s)} \\
&= \frac{d_{\pi_E}(s,a)}{d_{\pi_E}(s,a) + d_{\pi_G}(s,a)}
\end{aligned}
\tag{37}
$$

This is equal to the optimal discriminator of the GAN objective Eq. (16) when $x = (s,a)$. For this discriminator we showed in Section A.1 that the optimal generator $\pi_G^*$ is such that $d_{\pi_G^*}(s,a) = d_{\pi_E}(s,a) \ \forall (s,a) \in \mathcal{S} \times \mathcal{A}$, which is satisfied for $\pi_G^*(a|s) = \pi_E(a|s) \ \forall (s,a) \in \mathcal{S} \times \mathcal{A}$. Using the fact that

$$\sum_{a'} \exp \tilde{f}^*(s,a') = \sum_{a'} \pi_E(a'|s)\frac{d_{\pi_E}(s)}{d_{\pi_G}(s)} = \frac{d_{\pi_E}(s)}{d_{\pi_G}(s)}\sum_{a'}\pi_E(a'|s) = \frac{d_{\pi_E}(s)}{d_{\pi_G}(s)}. \tag{38}$$

we can combine Eq. (36) and Eq. (38) to write the expert's policy $\pi_E$ as a function of the optimal discriminator parameter $\tilde{f}^*$:

$$\pi_E(a|s) = \frac{\exp \tilde{f}^*(s,a)}{\sum_{a'}\exp \tilde{f}^*(s,a')} \quad \forall (s,a) \in \mathcal{S} \times \mathcal{A}. \tag{39}$$

This concludes the second part of the proof. $\qquad\square$

**Adversarial Soft-Q Fitting (ASQF) - practical algorithm** In a nutshell, Theorem 2 tells us that training the discriminator in Eq. (30) to distinguish between transitions from the expert and transitions from a generator policy can be seen as retrieving $\tilde{f}^*$ which plays the role of the expert's soft Q-function (i.e. which matches Eq. (1) for $\tilde{f}^* = \frac{1}{\alpha}Q^*_{\text{soft},E}$):

$$\pi_E(a|s) = \frac{\exp \tilde{f}^*(s,a)}{\sum_{a'}\exp \tilde{f}^*(s,a')} = \exp\left(\tilde{f}^*(s,a) - \log\sum_{a'}\exp \tilde{f}^*(s,a')\right), \tag{40}$$

Therefore, by training the discriminator, one simultaneously retrieves the optimal generator policy.

There is one caveat though: the summation over actions that is required in Eq. (40) to go from $\tilde{f}^*$ to the policy is intractable in continuous action spaces and would require an additional step such as a projection to a proper distribution (Haarnoja et al. [11] use a Gaussian) in order to draw samples and evaluate likelihoods. Updating in this way the generator policy to match a softmax over our learned state-action preferences ($\tilde{f}^*$) becomes very similar in requirements and computational load to a policy optimization step, thus defeating the purpose of this work which is to get rid of the policy optimization step. For this reason we only consider ASQF for discrete action spaces.

As explained in Section 3.3, in practice we optimize $D_{\tilde{f},\pi_G}$ only for a few steps before updating $\pi_G$ by normalizing $\exp \tilde{f}(s,a)$ over the action dimension. See Algorithm 2 for the pseudo-code.

---

**Algorithm 2:** Adversarial Soft-Q Fitting (ASQF)

---

**Require:** expert transitions $\mathcal{D}_E = \{(s_i, a_i)\}_{i=1}^{N_E}$
  Randomly initialize $\tilde{f}$ and get $\pi_G$ from Eq. (40)
  **for** steps $m = 0$ to $M$ **do**
    Collect transitions $\mathcal{D}_G = \{(s_i, a_i)\}_{i=1}^{N_G}$ by executing $\pi_G$
    Train $D_{\tilde{f},\pi_G}$ using binary cross-entropy on minibatches of transitions from $\mathcal{D}_E$ and $\mathcal{D}_G$
    Get $\pi_G$ from Eq. (40)
  **end for**

---

**Experimental results** Figure 4 shows that ASQF performs well on small scale environments but struggles and eventually fails on more complicated environments. Specifically, it seems that ASQF does not scale well with the observation space size. Indeed mountaincar, cartpole, lunarlander and pommerman have respectively an observation space dimensionality of 2, 4, 8 and 960. This may be due to the fact that the partition function Eq. (38) becomes more difficult to learn. Indeed, for each state, several transitions with different actions are required in order to learn it. Poorly approximating this partition function could lead to assigning too low a probability to expert-like actions and eventually failing to behave appropriately. ASAF on the other hand explicitly learns the probability of an action given the state – in other word it explicitly learns the partition function – and is therefore immune to that problem.

Figure 4: Comparison between ASAF-1 and ASQF, our two transition-wise methods, on environments with increasing observation space dimensionality

# C   Additional Experiments

## C.1   GAIL - Importance of Gradient Penalty

Figure 5: Comparison between original GAIL [13] and GAIL with gradient penalty (GP) [9, 14]

## C.2   Mimicking the expert

To ensure that our method actually mimics the expert and doesn't just learn a policy that collects high rewards when trained with expert demonstrations, we ran ASAF-1 on the Ant-v2 MuJoCo environment using various sets of 25 demonstrations. These demonstrations were generated from a Soft Actor-Critic agent at various levels of performance during its training. Since at low-levels of performance the variance of episode's return is high, we filtered collected demonstrations to lie in the targeted range of performance (e.g. return in [800, 1200] for the 1K set). Results in Figure 6 show that our algorithm succeeds at learning a policy that closely emulates various demonstrators (even when non-optimal).

Figure 6: ASAF-1 on Ant-v2. Colors are 1K, 2K, 3K, 4K, 5K expert's performance.

## C.3   Wall Clock Time

We report training times in Figure 7 and observe that ASAF-1 is always fastest to learn. Note however that reports of performance w.r.t wall-clock time should always be taken with a grain of salt as they are greatly influenced by hyper-parameters and implementation details.

Figure 7: Training times on MuJoCo tasks for 25 expert demonstrations.

# D Hyperparameter tuning and best configurations

## D.1 Classic Control

For this first set of experiments, we use the fixed hyperparameters presented in Table 1.

Table 1: Fixed Hyperparameters for classic control tasks

**RL COMPONENT**

| HYPER-PARAMETER | DISCRETE CONTROL | CONTINUOUS CONTROL |
|---|---|---|
| **SAC** | | |
| BATCH SIZE (IN TRANSITIONS) | 256 | 256 |
| REPLAY BUFFER LENGTH $|\mathcal{B}|$ | $10^6$ | $10^6$ |
| WARMUP (IN TRANSITIONS) | 1280 | 10240 |
| INITIAL ENTROPY WEIGHT $\alpha$ | 0.4 | 0.4 |
| GRADIENT NORM CLIPPING THRESHOLD | 0.2 | 1 |
| TRANSITIONS BETWEEN UPDATE | 40 | 1 |
| TARGET NETWORK WEIGHT $\tau$ | 0.01 | 0.01 |
| | | |
| **PPO** | | |
| BATCH SIZE (IN TRANSITIONS) | 256 | 256 |
| GAE PARAMETER $\lambda$ | 0.95 | 0.95 |
| TRANSITIONS BETWEEN UPDATE | - | 2000 |
| EPISODES BETWEEN UPDATES | 10 | - |
| EPOCHS PER UPDATE | 10 | 10 |
| UPDATE CLIPPING PARAMETER | 0.2 | 0.2 |

**REWARD LEARNING COMPONENT**

| HYPER-PARAMETER | DISCRETE CONTROL | CONTINUOUS CONTROL |
|---|---|---|
| **AIRL, GAIL, ASAF-1** | | |
| BATCH SIZE (IN TRANSITIONS) | 256 | 256 |
| TRANSITIONS BETWEEN UPDATE | - | 2000 |
| EPISODES BETWEEN UPDATES | 10 | - |
| EPOCHS PER UPDATE | 50 | 50 |
| GRADIENT VALUE CLIPPING THRESHOLD (ASAF-1) | - | 1 |
| | | |
| **ASAF, ASAF-$w$** | | |
| BATCH SIZE (IN TRAJECTORIES) | 10 | 10 |
| EPISODES BETWEEN UPDATES | 10 | 20 |
| EPOCHS PER UPDATE | 50 | 50 |
| WINDOW SIZE $w$ | (SEARCHED) | 200 |
| GRADIENT VALUE CLIPPING THRESHOLD | - | 1 |

For the most sensitive hyperparameters, the learning rates for the reinforcement learning and discriminator updates ($\epsilon_{\text{RL}}$ and $\epsilon_{\text{D}}$), we perform a random search over 50 configurations and 3 seeds each (for each algorithm on each task) for 500 episodes. We consider logarithmic ranges, i.e. $\epsilon = 10^u$ with $u \sim Uniform(-6, -1)$ for $\epsilon_{\text{D}}$ and $u \sim Uniform(-4, -1)$ for $\epsilon_{\text{RL}}$. We also include in this search the critic learning rate coefficient $\kappa$ for PPO also sampled according to a logarithmic scale with $u \sim Uniform(-2, 2)$ so that the effective learning rate for PPO's critic network is $\kappa \cdot \epsilon_{\text{RL}}$. For discrete action tasks, the window-size $w$ for ASAF-$w$ is sampled uniformly within $\{32, 64, 128\}$. The best configuration for each algorithm is presented in Tables 2 to 7. Figure 1 uses these configurations retrained on 10 seeds and twice as long.

Finally for all neural networks (policies and discriminators) for these experiments we use a fully-connected MLP with two hidden layers and ReLU activation (except for the last layer). We used hidden sizes of 64 for the discrete tasks and of 256 for the continuous tasks.

Table 2: Best found hyper-parameters for Cartpole

| HYPER-PARAMETER | ASAF | ASAF-$w$ | ASAF-1 | SQIL | AIRL + PPO | GAIL + PPO |
|---|---|---|---|---|---|---|
| DISCRIMINATOR UPDATE LR $\epsilon_D$ | 0.028 | 0.039 | 0.00046 | - | $2.5*10^{-6}$ | 0.00036 |
| RL UPDATE LR $\epsilon_{RL}$ | - | - | - | 0.0067 | 0.0052 | 0.012 |
| CRITIC LR COEFFICIENT $\kappa$ | - | - | - | - | 0.25 | 0.29 |
| WINDOW SIZE $w$ | - | 64 | 1 | - | - | - |
| WINDOW STRIDE | - | 64 | 1 | - | - | - |

Table 3: Best found hyper-parameters for Mountaincar

| HYPER-PARAMETER | ASAF | ASAF-$w$ | ASAF-1 | SQIL | AIRL + PPO | GAIL + PPO |
|---|---|---|---|---|---|---|
| DISCRIMINATOR UPDATE LR $\epsilon_D$ | 0.059 | 0.059 | 0.0088 | - | 0.0042 | 0.00016 |
| RL UPDATE LR $\epsilon_{RL}$ | - | - | - | 0.062 | 0.016 | 0.0022 |
| CRITIC LR COEFFICIENT $\kappa$ | - | - | - | - | 4.6 | 0.018 |
| WINDOW SIZE $w$ | - | 32 | 1 | - | - | - |
| WINDOW STRIDE | - | 32 | 1 | - | - | - |

Table 4: Best found hyper-parameters for Lunarlander

| HYPER-PARAMETER | ASAF | ASAF-$w$ | ASAF-1 | SQIL | AIRL + PPO | GAIL + PPO |
|---|---|---|---|---|---|---|
| DISCRIMINATOR UPDATE LR $\epsilon_D$ | 0.0055 | 0.0015 | 0.00045 | - | 0.0002 | 0.00019 |
| RL UPDATE LR $\epsilon_{RL}$ | - | - | - | 0.0036 | 0.0012 | 0.0016 |
| CRITIC LR COEFFICIENT $\kappa$ | - | - | - | - | 0.48 | 8.5 |
| WINDOW SIZE $w$ | - | 32 | 1 | - | - | - |
| WINDOW STRIDE | - | 32 | 1 | - | - | - |

Table 5: Best found hyper-parameters for Pendulum

| HYPER-PARAMETER | ASAF | ASAF-$w$ | ASAF-1 | SQIL | AIRL + PPO | GAIL + PPO |
|---|---|---|---|---|---|---|
| DISCRIMINATOR UPDATE LR $\epsilon_D$ | 0.00069 | 0.00082 | 0.00046 | - | $4.3*10^{-6}$ | $1.6*10^{-5}$ |
| RL UPDATE LR $\epsilon_{RL}$ | - | - | - | 0.0001 | 0.00038 | 0.00028 |
| CRITIC LR COEFFICIENT $\kappa$ | - | - | - | - | 0.028 | 84 |
| WINDOW SIZE $w$ | - | 200 | 1 | - | - | - |
| WINDOW STRIDE | - | 200 | 1 | - | - | - |

Table 6: Best found hyper-parameters for Mountaincar-c

| HYPER-PARAMETER | ASAF | ASAF-$w$ | ASAF-1 | SQIL | AIRL + PPO | GAIL + PPO |
|---|---|---|---|---|---|---|
| DISCRIMINATOR UPDATE LR $\epsilon_D$ | 0.00021 | $3.8*10^{-5}$ | $6.2*10^{-6}$ | - | $1.7*10^{-5}$ | $1.5*10^{-5}$ |
| RL UPDATE LR $\epsilon_{RL}$ | - | - | - | 0.0079 | 0.0012 | 0.0052 |
| CRITIC LR COEFFICIENT $\kappa$ | - | - | - | - | 10 | 12 |
| WINDOW SIZE $w$ | - | 200 | 1 | - | - | - |
| WINDOW STRIDE | - | 200 | 1 | - | - | - |

Table 7: Best found hyper-parameters for Lunarlander-c

| HYPER-PARAMETER | ASAF | ASAF-$w$ | ASAF-1 | SQIL | AIRL + PPO | GAIL + PPO |
|---|---|---|---|---|---|---|
| DISCRIMINATOR UPDATE LR $\epsilon_D$ | 0.0051 | 0.0022 | 0.0003 | - | 0.0045 | 0.00014 |
| RL UPDATE LR $\epsilon_{RL}$ | - | - | - | 0.0027 | 0.00031 | 0.00049 |
| CRITIC LR COEFFICIENT $\kappa$ | - | - | - | - | 14 | 0.01 |
| WINDOW SIZE $w$ | - | 200 | - | - | - | - |
| WINDOW STRIDE | - | 200 | - | - | - | - |

## D.2 MuJoCo

For MuJoCo experiments (Hopper-v2, Walker2d-v2, HalfCheetah-v2, Ant-v2), the fixed hyperparameters are presented in Table 8. For all exeperiments, fully-connected MLPs with two hidden layers and ReLU activation (except for the last layer) were used, where the number of hidden units is equal to 256.

Table 8: Fixed hyperparameters for MuJoCo environments.

| RL COMPONENT | |
| --- | --- |
| HYPER-PARAMETER | HOPPER, WALKER2D, HALFCHEETAH, ANT |
| | |
| **PPO (FOR GAIL)** | |
| GAE PARAMETER $\lambda$ | 0.98 |
| TRANSITIONS BETWEEN UPDATES | 2000 |
| EPOCHS PER UPDATE | 5 |
| UPDATE CLIPPING PARAMETER | 0.2 |
| CRITIC LR COEFFICIENT $\kappa$ | 0.25 |
| DISCOUNT FACTOR $\gamma$ | 0.99 |
| **REWARD LEARNING COMPONENT** | |
| HYPER-PARAMETER | HOPPER, WALKER2D, HALFCHEETAH, ANT |
| | |
| **GAIL** | |
| TRANSITIONS BETWEEN UPDATES | 2000 |
| | |
| **ASAF** | |
| EPISODES BETWEEN UPDATES | 25 |
| | |
| **ASAF-1 AND ASAF-*w*** | |
| TRANSITIONS BETWEEN UPDATES | 2000 |

For SQIL we used SAC with the same hyperparameters that were used to generate the expert demonstrations. For ASAF, ASAF-1 and ASAF-*w*, we set the learning rate for the discriminator at 0.001 and ran random searches over 25 randomly sampled configurations and 2 seeds for each task to select the other hyperparameters for the discriminator training. These hyperparameters included the discriminator batch size sampled from a uniform distribution over $\{10, 20, 30\}$ for ASAF and ASAF-*w* (in trajectories) and over $\{100, 500, 1000, 2000\}$ for ASAF-1 (in transitions), the number of epochs per update sampled from a uniform distribution over $\{10, 20, 50\}$, the gradient norm clipping threshold sampled form a uniform distribution over $\{1, 10\}$, the window-size (for ASAF-*w*) sampled from a uniform distribution over $\{100, 200, 500, 1000\}$ and the window stride (for ASAF-*w*) sampled from a uniform distribution over $\{1, 50, w\}$. For GAIL, we obtained poor results using the original hyperparameters from [13] for a number of tasks so we ran random searches over 100 randomly sampled configurations for each task and 2 seeds to select for the following hyperparameters: the log learning rate of the RL update and the discriminator update separately sampled from uniform distributions over $[-7, -1]$, the gradient norm clipping for the RL update and the discriminator update separately sampled from uniform distributions over $\{None, 1, 10\}$, the number of epochs per update sampled from a uniform distribution over $\{5, 10, 30, 50\}$, the gradient penalty coefficient sampled from a uniform distribution over $\{1, 10\}$ and the batch size for the RL update and discriminator update separately sampled from uniform distributions over $\{100, 200, 500, 1000, 2000\}$.

#### Table 9: Best found hyper-parameters for the Hopper-v2 environment

| HYPER-PARAMETER | ASAF | ASAF-*w* | ASAF-1 | SQIL | GAIL + PPO |
|---|---|---|---|---|---|
| RL BATCH SIZE (IN TRANSITIONS) | - | - | - | 256 | 200 |
| DISCRIMINATOR BATCH SIZE (IN TRANSITIONS) | - | - | 100 | - | 2000 |
| DISCRIMINATOR BATCH SIZE (IN TRAJECTORIES) | 10 | 10 | - | - | - |
| GRADIENT CLIPPING (RL UPDATE) | - | - | - | - | 1. |
| GRADIENT CLIPPING (DISCRIMINATOR UPDATE) | 10. | 10. | 1. | - | 1. |
| EPOCHS PER UPDATE | 50 | 50 | 30 | - | 5 |
| GRADIENT PENALTY (DISCRIMINATOR UPDATE) | - | - | - | - | 1. |
| RL UPDATE LR $\epsilon_{\text{RL}}$ | - | - | - | $3*10^{-4}$ | $1.8*10^{-5}$ |
| DISCRIMINATOR UPDATE LR $\epsilon_{\text{D}}$ | 0.001 | 0.001 | 0.001 | - | 0.011 |
| WINDOW SIZE *w* | - | 200 | 1 | - | - |
| WINDOW STRIDE | - | 1 | 1 | - | - |

#### Table 10: Best found hyper-parameters for the HalfCheetah-v2 environment

| HYPER-PARAMETER | ASAF | ASAF-*w* | ASAF-1 | SQIL | GAIL + PPO |
|---|---|---|---|---|---|
| RL BATCH SIZE (IN TRANSITIONS) | - | - | - | 256 | 1000 |
| DISCRIMINATOR BATCH SIZE (IN TRANSITIONS) | - | - | 100 | - | 100 |
| DISCRIMINATOR BATCH SIZE (IN TRAJECTORIES) | 10 | 10 | - | - | - |
| GRADIENT CLIPPING (RL UPDATE) | - | - | - | - | - |
| GRADIENT CLIPPING (DISCRIMINATOR UPDATE) | 10. | 1 | 1 | - | 10 |
| EPOCHS PER UPDATE | 50 | 10 | 30 | - | 30 |
| GRADIENT PENALTY (DISCRIMINATOR UPDATE) | - | - | - | - | 1. |
| RL UPDATE LR $\epsilon_{\text{RL}}$ | - | - | - | $3*10^{-4}$ | 0.0006 |
| DISCRIMINATOR UPDATE LR $\epsilon_{\text{D}}$ | 0.001 | 0.001 | 0.001 | - | 0.023 |
| WINDOW SIZE *w* | - | 200 | 1 | - | - |
| WINDOW STRIDE | - | 1 | 1 | - | - |

#### Table 11: Best found hyper-parameters for the Walker2d-v2 environment

| HYPER-PARAMETER | ASAF | ASAF-*w* | ASAF-1 | SQIL | GAIL + PPO |
|---|---|---|---|---|---|
| RL BATCH SIZE (IN TRANSITIONS) | - | - | - | 256 | 200 |
| DISCRIMINATOR BATCH SIZE (IN TRANSITIONS) | - | - | 500 | - | 2000 |
| DISCRIMINATOR BATCH SIZE (IN TRAJECTORIES) | 20 | 20 | - | - | - |
| GRADIENT CLIPPING (RL UPDATE) | - | - | - | - | - |
| GRADIENT CLIPPING (DISCRIMINATOR UPDATE) | 10. | 1. | 10. | - | - |
| EPOCHS PER UPDATE | 30 | 10 | 50 | - | 30 |
| GRADIENT PENALTY (DISCRIMINATOR UPDATE) | - | - | - | - | 1. |
| RL UPDATE LR $\epsilon_{\text{RL}}$ | - | - | - | $3*10^{-4}$ | 0.00039 |
| DISCRIMINATOR UPDATE LR $\epsilon_{\text{D}}$ | 0.001 | 0.001 | 0.001 | - | 0.00066 |
| WINDOW SIZE *w* | - | 100 | 1 | - | - |
| WINDOW STRIDE | - | 1 | 1 | - | - |

#### Table 12: Best found hyper-parameters for the Ant-v2 environment

| HYPER-PARAMETER | ASAF | ASAF-*w* | ASAF-1 | SQIL | GAIL + PPO |
|---|---|---|---|---|---|
| RL BATCH SIZE (IN TRANSITIONS) | - | - | - | 256 | 500 |
| DISCRIMINATOR BATCH SIZE (IN TRANSITIONS) | - | - | 100 | - | 100 |
| DISCRIMINATOR BATCH SIZE (IN TRAJECTORIES) | 20 | 20 | - | - | - |
| GRADIENT CLIPPING (RL UPDATE) | - | - | - | - | - |
| GRADIENT CLIPPING (DISCRIMINATOR UPDATE) | 10. | 1. | 1. | - | 10. |
| EPOCHS PER UPDATE | 50 | 50 | 10 | - | 50 |
| GRADIENT PENALTY (DISCRIMINATOR UPDATE) | - | - | - | - | 10 |
| RL UPDATE LR $\epsilon_{\text{RL}}$ | - | - | - | $3*10^{-4}$ | $8.5*10^{-5}$ |
| DISCRIMINATOR UPDATE LR $\epsilon_{\text{D}}$ | 0.001 | 0.001 | 0.001 | - | 0.0016 |
| WINDOW SIZE *w* | - | 200 | 1 | - | - |
| WINDOW STRIDE | - | 50 | 1 | - | - |

### D.3 Pommerman

For this set of experiments, we use a number of fixed hyperparameters for all algorithms either inspired from their original papers for the baselines or selected through preliminary searches. These fixed hyperparameters are presented in Table 13.

Table 13: Fixed Hyperparameters for Pommerman Random-Tag environment.

**RL COMPONENT**

| HYPER-PARAMETER | POMMERMAN RANDOM-TAG |
|---|---|
| **SAC** | |
| BATCH SIZE (IN TRANSITIONS) | 256 |
| REPLAY BUFFER LENGTH $\|\mathcal{B}\|$ | $10^5$ |
| WARMUP (IN TRANSITIONS) | 1280 |
| INITIAL ENTROPY WEIGHT $\alpha$ | 0.4 |
| GRADIENT NORM CLIPPING THRESHOLD | 0.2 |
| TRANSITIONS BETWEEN UPDATE | 10 |
| TARGET NETWORK WEIGHT $\tau$ | 0.05 |
| | |
| **PPO** | |
| BATCH SIZE (IN TRANSITIONS) | 256 |
| GAE PARAMETER $\lambda$ | 0.95 |
| EPISODES BETWEEN UPDATES | 10 |
| EPOCHS PER UPDATE | 10 |
| UPDATE CLIPPING PARAMETER | 0.2 |
| CRITIC LR COEFFICIENT $\kappa$ | 0.5 |

**REWARD LEARNING COMPONENT**

| HYPER-PARAMETER | POMMERMAN RANDOM-TAG |
|---|---|
| **AIRL, GAIL, ASAF-1** | |
| BATCH SIZE (IN TRANSITIONS) | 256 |
| EPISODES BETWEEN UPDATES | 10 |
| EPOCHS PER UPDATE | 10 |
| | |
| **ASAF, ASAF-*w*** | |
| BATCH SIZE (IN TRAJECTORIES) | 5 |
| EPISODES BETWEEN UPDATES | 10 |
| EPOCHS PER UPDATE | 10 |

For the most sensitive hyperparameters, the learning rates for the reinforcement learning and discriminator updates ($\epsilon_{\text{RL}}$ and $\epsilon_{\text{D}}$), we perform a random search over 25 configurations and 2 seeds each for all algorithms. We consider logarithmic ranges, i.e. $\epsilon = 10^u$ with $u \sim Uniform(-7, -3)$ for $\epsilon_{\text{D}}$ and $u \sim Uniform(-4, -1)$ for $\epsilon_{\text{RL}}$. We also include in this search the window-size $w$ for ASAF-*w*, sampled uniformly within $\{32, 64, 128\}$. The best configuration for each algorithm is presented in Table 14. Figure 3 uses these configurations retrained on 10 seeds.

Table 14: Best found hyper-parameters for the Pommerman Random-Tag environment

| HYPER-PARAMETER | ASAF | ASAF-*w* | ASAF-1 | SQIL | AIRL + PPO | GAIL + PPO | BC |
|---|---|---|---|---|---|---|---|
| DISCRIMINATOR UPDATE LR $\epsilon_{\text{D}}$ | 0.0007 | 0.0002 | 0.0001 | - | $3.1*10^{-7}$ | $9.3*10^{-7}$ | 0.00022 |
| RL UPDATE LR $\epsilon_{\text{RL}}$ | - | - | - | 0.00019 | 0.00017 | 0.00015 | - |
| WINDOW SIZE $w$ | - | 32 | 1 | - | - | - | - |
| WINDOW STRIDE | - | 32 | 1 | - | - | - | - |

Finally for all neural networks (policies and discriminators) we use the same architecture. Specifically, we first process the feature maps (see Section E.3) using a 3-layers convolutional network with number of hidden feature maps of 16, 32 and 64 respectively. Each one of these layers use a kernel size of 3x3 with stride of 1, no padding and a ReLU activation. This module ends with a fully connected layer of hidden size 64 followed by a ReLU activation. The output vector is then concatenated to the unprocessed additional information vector (see Section E.3) and passed through a final MLP with two hidden layers of size 64 and ReLU activations (except for the last layer).

# E Environments and expert data

## E.1 Classic Control

The environments used here are the reference Gym implementations for classic control[2] and for Box2D[3]. We generated the expert trajectories for mountaincar (both discrete and continuous version) by hand using keyboard inputs. For the other tasks, we trained our SAC implementation to get experts on the discrete action tasks and our PPO implementation to get experts on the continuous action tasks.

## E.2 MuJoCo

The experts were trained using our implementation of SAC [11] the state-of-the-art RL algorithm in MuJoCo continuous control tasks. Our implementation basically refactors the SAC implementation from Rlpyt[4]. We trained SAC agent for 1,000,000 steps for Hopper-v2 and 3,000,000 steps for Walker2d-v2 and HalfCheetah-v2 and Ant-v2. We used the default hyper-parameters from Rlpyt.

## E.3 Pommerman

The observation space that we use for Pommerman domain [21] is composed of a set of 15 feature maps as well as an additional information vector. The feature maps whose dimensions are given by the size of the board (8x8 in the case of 1vs1 tasks) are one-hot across the third dimension and represent which element is present at which location. Specifically, these feature maps identify whether a given location is the current player, an ally, an ennemy, a passage, a wall, a wood, a bomb, a flame, fog, a power-up. Other feature maps contain integers indicating bomb blast stength, bomb life, bomb moving direction and flame life for each location. Finally, the additional information vecor contains the time-step, number of ammunition, whether the player can kick and blast strengh for the current player. The agent has an action space composed of six actions: do-nothing, up, down, left, right and lay bomb.

For these experiments, we generate the expert demonstrations using Agent47Agent, the open-source champion algorithm of the FFA 2018 competition [28] which uses hardcoded heuristics and Monte-Carlo Tree-Search[5]. While this agent occasionally eliminates itself during a match, we only select trajectories leading to a win as being expert demonstrations.

## E.4 Demonstrations summary

Table 15 provides a summary of the expert data used.

Table 15: Expert demonstrations used for Imitation Learning

| TASK-NAME | EXPERT MEAN RETURN | NUMBER OF EXPERT TRAJECTORIES |
|---|---|---|
| CARTPOLE | 200.0 | 10 |
| MOUNTAINCAR | -108.0 | 10 |
| LUNARLANDER | 277.5 | 10 |
| PENDULUM | -158.6 | 10 |
| MOUNTAINCAR-C | 93.92 | 10 |
| LUNARLANDER-C | 266.1 | 10 |
| HOPPER | 3537 | 25 |
| WALKER2D | 5434 | 25 |
| HALFCHEETAH | 7841 | 25 |
| ANT | 5776 | 25 |
| POMMERMAN RANDOM-TAG | 1 | 300, 150, 75, 15, 5, 1 |

## Footnotes

[2]See: http://gym.openai.com/envs/#classic_control

[3]See: http://gym.openai.com/envs/#box2d

[4]See: https://github.com/astooke/rlpyt

[5]See: https://github.com/YichenGong/Agent47Agent/tree/master/pommerman