[Reviews · NeurIPS 2020]

Review 1

Summary and Contributions: This paper extends adversarial imitation learning by learning a structured discriminator. As in GAIL, a discriminator is learned to differentiate between expert and policy behavior and then utilized as a reward function to improve the policy. In this work, the discriminator is formulated to naturally include the generator / policy resolving the need for an additional policy optimization step. This reduces the computational cost of learning a policy from adversarial imitation by reducing it to only learning the discriminator. This method performs better than other common techniques from a limited number of demonstrations

Strengths: The paper has a strong set of theoretical and experimental results on relevant problems to the NeurIPS community. The paper addresses the limitation of adversarial imitation learning, a core problem, and compares against a common solution with better results. The idea of re-using a component of the discriminator as a parameterization of the policy while not entirely novel (see weaknesses) is explored here with competitive results and the ability to learn policies from adversarial imitation without direct policy optimization would be very useful. As mentioned, this would significantly reduce the computational cost of adversarial imitation methods and stabilize a generally unstable method. The experiments are run on a variety of common benchmarks and while not significantly improved more baselines demonstrate stable training. The results are also impressive with a limited number of demonstrations. Additionally, the authors have relatively well organized and understandable code to reproduce their results.

Weaknesses: The work, while useful, relies heavily on previous work Finn et al and Fu et al which illustrate the particular forms of the discriminator which can be used to improve adversarial training. I think further discussion on this point is warranted to at least clarify how this work improves on the existing literature. A few other points: There seems to be a large gain from the windowed approach, but it seems like this technique would also be valid for GAIL or AIRL? It’s not clear to me why AIRL is dropped from the Mujoco benchmarks Gradient clipping is introduced to improve GAIL, but perhaps something like https://arxiv.org/abs/1810.00821 should also be benchmarked against to improve stability

Correctness: Yes, both the claims and method seem correct

Clarity: I found the paper to be generally well written and clear.

Relation to Prior Work: One minor point is that is seems like reference [13] Ho and Ermon while referenced throughout is left out of the related works section Additionally, as noted above, I think more is needed to differentiate this work from Fu et al [6]. The discriminator form as noted is similar and Fu et al also note in their appendix that the discriminator recovers the advantage function if parameterized correctly. However, it does seem to be that this work directly transforms the advantage function into the policy skipping the policy optimization step all together. Additionally, while not directly related, I do notice quite a bit of similarities here with recent work on re-casting discriminators as energy-based models. Some exploration of the relation there may also be useful. https://arxiv.org/abs/1912.03263 https://arxiv.org/abs/2004.09395

Reproducibility: Yes

Additional Feedback:


Review 2

Summary and Contributions: AUTHOR RESPONSE: Thanks for the detailed response! The rebuttal addressed most of my concerns; thanks for running the experiment with expert data of varying quality. I've increased my review (7 -> 8). This paper proposes an imitation learning method that doesn't require running RL in the inner loop. The main idea is to adopt the adversarial IRL framework to use a discriminator that is defined implicitly in terms of a policy. Thus, optimizing the discriminator (via classification) results in learning the policy. The paper provides some analysis that this method recovers the expert policy, and includes experiments showing that it outperforms prior methods.

Strengths: To the best of my knowledge, the proposed algorithm is novel. It's simplicity and performance (at least on the reported experiments) bodes well for adoption into the NeurIPS community. The experimental evaluation is fairly thorough.

Weaknesses: * The experimental evaluation omits some more recent baselines, such as behavior cloning (in Fig 1, 2), AIRL (in Fig 2), [Ghasemipour 19], and [Kostrikov 20]. * Some of the claims in the related works section seem questionable (see relation w/ prior work) * The writing could be clearer in some places (see minor comments below)

Correctness: The claims and experiments seem mostly correct. While the analysis shows that the solution to the min-max problem (Eq. 10) is the expert policy, I'm unsure whether the proposed proceduring (which solves the inner max problem, and then sets the outer policy equal to the inner policy) is guaranteed to converge. I would increase my review if the paper were updated to include a proof that the proposed algorithm converges. One comment about the experiments is that they don't actually show that the proposed method mimics the expert, only that running the proposed algorithm with data generated from an expert results in high reward. I would increase my review if an experiment were added to show that the learned policy actually mimics the demonstrator. One potential experiment would take the half-cheetah domain and generate demonstration datasets with rewards of 1k, 2k, 3k, 4k, ... Running the proposed method on each of these datasets should achieve rewards of 1k, 2k, 3k, 4k. One minor nitpick are the claims in the abstract that the method is less "delicate" and more computationally efficient than prior methods. While it seems likely that these claims are true, I'm sure if they're actually substantiated in the experiments/appendix.

Clarity: The writing is mostly clear. I found section 3 slightly confusing because the optimization problem (Eq 13) is written in terms of the discriminator, which in turn depends on the learned policy or Q function. Substituting the definition of D into Eq 3 might help clarify this.

Relation to Prior Work: Mostly, though I'd recommend adding some discussion of [Ghasemipour 19]. I found some of the claims in Section 4 a bit overzealous. For example, I don't think that the claim that methods that "abandon the MaxEnt IRL framework" necessarily "fall back into the policy ambiguity problem." Similarly, the problem discussed in L212 seems prevalent in all MaxEnt IRL methods (not just SQIL), and seems like a good thing: when we've converged to the optimal policy, we don't want to provide a reward that changes the policy.

Reproducibility: Yes

Additional Feedback: Minor comments: * "Imitation Learning" isn't capitalized. * L22 "this" ambiguous pronoun reference * L23 "struggle to generalize" add a citation * L29 - 33 "Early methods ... Recent approaches...": I think the important difference between these methods is *not* whether they run RL to convergence, but rather that recent approaches use function approximators to scale to large tasks. Indeed, I believe that the convergence guarantees for recent approaches only hold if they ran RL to convergence in the inner loop. * L35 "it has" -> "they have" * L38"handcrafting": It seems disingenuous to say that r(s, a) = 1 is handcrafted... * L62: "]0, 1[" -> "[0, 1]" * L69: I think you can cut "if \gamma ... and T < \infty." The summation is correct when \gamma = 1. * Eq 2, 3: Remove numbers from equations that are not referred to later (e.g., with \nonumber or \begin{equation*}) * L114 "arbitrary distribution \tilde{p} that can both ..." This makes it sound like \tilde{p} will satisfy two properties, rather than that p_G and \tilde{p} both satisfy one property. * Fig 3 (right): Where is this figure discussed in the main text? Also, why is the X axis reversed? * L301 "rewards along" -> "rewards, along"


Review 3

Summary and Contributions: Thank you for your thorough comments. The new experimental results and details are convincing. --- This paper proposes a reformulation of imitation learning such that it avoids the complex back and forth GAN optimization that is difficult to tune correctly. Instead, the authors show that the discriminator and generator can be both trained simultaneously via a structured discriminator. Using a simple binary cross entropy loss enables learning the discriminator as well as the generator via supervised learning without any need for running RL. The proposed approach works well across a variety of benchmarks and appears to be a very promising approach to imitation learning that is much simpler and outperforms prior work.

Strengths: The paper has a nice blend of theory and rigorous experimental results. It is well written and addresses the difficultly in getting GAN-style methods like GAIL to work. The experimental results are very thorough and nicely demonstrate the benefits of the proposed approach.

Weaknesses: The proposed approach bears strong resemblance to behavioral cloning, but the approach isn't compared to behavioral cloning in the experiments and no intuition or theory is given for how it differs from standard behavioral cloning. In particular, the ASAF-1 algorithm seems very similar to BC. This method does not recover a reward function as many other methods do. This is problematic if the goal is to explain the demonstrator's behavior, transfer to a similar task with different dynamics, or provide safety bounds on performance. The proposed approach requires demonstrator actions and it is unclear how it could be extended to work purely from observations as many recent imitation learning approaches do.

Correctness: The claims and empirical methodology are nicely supported.

Clarity: The paper is very well written

Relation to Prior Work: The paper only mentions the work of Reddy et al. when talking about recent approaches to make imitation learning simpler and more efficient. There are several other recent papers that seek the same goal: Uchibe. "Model-free deep inverse reinforcement learning by logistic regression." Neural Processing Letters, 2018. Brown et al. "Better-than-Demonstrator Imitation Learning via Automatically-Ranked Demonstrations." CoRL, 2019. Brantley et al. "Disagreement-regularized imitation learning." ICRL, 2019. Each of these works seeks to make imitation learning more tractable by turning reward learning into supervised learning and avoiding the unstable minmax GAN training that this paper also seeks to avoid. Uchibe and Brown et al. both use a logistic/binary cross entropy loss to train a reward function which bears resemblance to the proposed approach which trains the policy's state-conditioned action distributions via a cross-entropy loss. Brantley et al. propose an approach very similar to Reddy et al. to recover a simple reward function from demonstrations. All of these methods require RL so they do not detract from the contributions of this work, but would make the related work discussion more complete.

Reproducibility: Yes

Additional Feedback: The paper mentions that they use the MaxEnt policy class shown in Equation (2) to train the policy, but then state that the policy is trained to minimize Eq (13). Are value functions also learned during training? How is the entropy regularization included in minimizing equation (13)? It seems that this minimization just requires a parameterized policy that can be trained via gradient descent so it does not appear to require the max causal entropy policy assumption and it is unclear how to train such a policy since it depends on the reward function which is not being recovered in the proposed approach.


Review 4

Summary and Contributions: The paper proposes an imitation learning algorithm based on the GAIL framework. It applies a structured discriminator of GAN to the imitation learning problem. Using the structured discriminator, the policy optimization step can be avoided since the optimal policy is recovered as a byproduct from the discriminator learning step. The paper describes the algorithm and simulation results using classic control problems and three MuJoCo tasks and the Pommerman game.

Strengths: The proposed method can eliminate the policy optimization step required in GAIL. It can potentially make IRL practical for a wide range of real applications.

Weaknesses: While the paper proposes an interesting variant of GAIL without the policy optimization step, the simulation results provided in the paper are weak. (1) The paper lacks empirical study of the effect of the structured discriminator: its convergence, its efficiency over GAIL, and stability during training compared to GAIL, to name a few. (2) The paper only reports 3 tasks from MuJoCo and the performance of the proposed method is not always better than GAIL. In addition, other methods, such as AIRL and SQIL, are not compared for the MuJoCo experiments. It makes the reader to speculate whether the proposed method performs better than existing approaches for discrete or simpler problems only. The paper does not report the total training times (in seconds or FLOPs) so it is not possible to determine whether the proposed method can shorten the training time over existing methods. It should discuss the time saved from eliminating the RL step compared to the time required to learn the more complex structured discriminator.

Correctness: The supplementary material (zip file) cannot be opened so I couldn’t verify materials in the supplementary.

Clarity: Overall, good.

Relation to Prior Work: Yes.

Reproducibility: Yes

Additional Feedback:

[Author Response · NeurIPS 2020]

We would like to thank the reviewers for their thorough evaluations and for bringing to our attention some missing
citations and typos, these will be corrected in the updated manuscript. We answer here specific questions raised by the
reviewers and present requested additional experiments.

***Additional MuJoCo experiments (R1, R2, R4).*** We focus on GAIL because AIRL claims to perform on par with GAIL
on MuJoCo. We present in Figure 2 below additional results on the MuJoCo envs (as well as the additional Ant env)
where GAIL has been re-tuned for further improvement and SQIL and ASAF-1 have been added. We see that even with
careful tuning GAIL is outperformed by our method and that SQIL's instability is exacerbated on MuJoCo.

We also ran ASAF-1 on the Ant-v2 MuJoCo environment using various sets
of 25 demonstrations ***(as requested by R2)***. These demonstrations were
generated from a Soft Actor-Critic agent at various levels of performance
during its training. Since at low-levels of performance the variance of
episode return is high, we filtered collected demonstrations to lie in the
targeted range of performance (e.g. return in [800, 1200] for the 1K set).
Results in Figure 1 show that our algorithm succeeds at learning a policy
that closely emulates various demonstrators (even when non-optimal).

Figure 1: ASAF-1 on Ant-v2. Colors are
1K, 2K, 3K, 4K, 5K expert's performance.

***Comparison of our methods vs Finn et al. [1] and Fu et al. [2] and***
***Behavioral Cloning (BC) (R1, R3).*** Our work builds on [1], yet its novelty
is to explicitly express the probability of a trajectory in terms of the policy in
order to directly learn this latter when training the discriminator. In contrast, [2] considers a transition-wise discriminator
with un-normalized probabilities which makes is closer to ASQF (Appendix B) than to ASAF-1. Additionally, AIRL
[2] minimizes the Kullback-Leiber Divergence [3] between occupancy measures whereas ASAF minimizes the Jensen-
Shanon Divergence between trajectories likelihood. Finally, BC uses the loss function from supervised learning
(classification or regression) to match expert's actions given expert's states and suffers from compounding error due
to co-variate shift [4] since it only learns on the expert state-action visitations (demonstrations) without environment
interaction. Contrarily, ASAF-1 uses the binary cross entropy loss in Eq. (13) and does not suffer from compounding
error as it learns on both generated and expert's trajectories.

***Can windowed approach be used for GAIL and AIRL? (R1).*** GAIL's and AIRL's discriminators are updated based on
transitions so that a step-wise reward can be learned and used in RL loops. Therefore, the windowed approach, which
suits trajectory-wise formulations, could not be applied without major modification of their formulations.

***Reward Acquisition from ASAF (R3).*** Although ASAF does not explicitly acquire reward during training, we can
retrieve step-wise soft advantages $\log \pi(a|s)$ from learned agent's policy $\pi(a|s)$ which can be used as a reward
function [2, 5].

***Training Time (R4).*** Due to lack of room we cannot add here the equivalent of Figure 2 with wall clock time as x-axis
but we will add it to the updated manuscript. Our observation is that ASAF-1 is always fastest to learn, e.g., 361.2s
(ASAF-1), 473.1s (SQIL), 1561.0s (ASAF), 1763.6s (GAIL), 2079.1s (ASAF-w) were taken to reach a performance of
2000 in Hopper environment. Note however that reports of performance w.r.t wall-clock time should always be taken
with a grain of salt as they are greatly influenced by hyper-parameters and implementation details.

***No entropy regularization in loss (R3).*** We learn in the softmax policy class of Eq. (2) since it contains the expert's
policy (See Section 3.2) but optimize Eq. (13) without entropy regularization as was done in the GAIL paper.

***Additional Concerns. (To R3)*** We will update our related works with your recommendations. ***(To R4)*** It seems that our
supplementary files have been correctly uploaded since R1 was able to read through our code. We are sorry that you
couldn't access it. Unfortunately, we are not allowed to put an external link in this rebuttal. We suggest that you contact
the AC about this issue.

***References.*** [1] Finn et al., "A connection between generative ...," (2016) [2] Fu et al., "Learning robust rewards ...,"
(2017) [3] Ghasemipour et al., "A divergence minimization perspective ...," (2019) [4] Ross and Bagnell, "Efficient
reductions for imitation learning," (2010) [5] Schulman et al., "High-dimensional continuous control ...," (2015)

Figure 2: Results on Mujoco environments with improved tuning for GAIL, added SQIL and ASAF-1 and env (Ant)

[Meta-Review · NeurIPS 2020]

Even before the author response, the reviewers agreed that the results and approach were interesting. The response addressed the reviewers remaining concerns about novelty, baseline strength, and positioning with respect to prior work. This led the reviewers to a consensus that the paper should be accepted.